# *Native Chinese Reader*: A Dataset Towards Native-Level Chinese Machine Reading Comprehension

**Shusheng Xu**[1]*, **Yichen Liu**[2,5]*, **Xiaoyu Yi**[3,5], **Siyuan Zhou**[4,5], **Huizi Li**[5], **Yi Wu**[1]

[1] IIIS, Tsinghua University, [2] New York University [3] Shenzhen University,
[4] Peking University, [5] Haihua Institute for Frontier Information Technology

xuss20@mails.tsinghua.edu.cn, yl7043@nyu.edu, yixiaoyuszu@outlook.com
elegyhunter@gmail.com, lhz@haihua.org.cn, jxwuyi@gmail.com

## Abstract

We present *Native Chinese Reader (NCR)*, a new machine reading comprehension (MRC) dataset with particularly long articles in both *modern* and *classical* Chinese. NCR is collected from the exam questions for the Chinese course in China's high schools, which are designed to evaluate the language proficiency of native Chinese youth. Existing Chinese MRC datasets are either domain-specific or focusing on short contexts of a few hundreds of characters in modern Chinese only. By contrast, NCR contains 8390 documents with an average length of 1024 characters covering a wide range of Chinese writing styles, including modern articles, classical literature and classical poetry. A total of 20477 questions on these documents also require strong reasoning abilities and common sense to figure out the correct answers. We implemented multiple baseline models using popular Chinese pretrained models and additionally launched an online competition using our dataset to examine the limit of current methods. The best model achieves 59% test accuracy while human evaluation shows an average accuracy of 79%, which indicates a significant performance gap between current MRC models and native Chinese speakers. We release the dataset at https://sites.google.com/view/native-chinese-reader/.

## 1 Introduction

Machine reading comprehension (MRC) is one of the fundamental tasks in natural language understanding, which requires a machine to read a document to correctly answer questions based on the context. MRC has attracted significant efforts from both academia and industry with continuous development of MRC datasets [15, 23, 19, 26, 28, 33], which also keeps pushing the frontier of MRC models and learning algorithms [11, 22, 20, 3] to eventually bridge the gap between AI systems and human readers.

In addition to the advances in English MRC, researchers have also made substantial progresses in Chinese MRC challenges with many high-quality Chinese MRC datasets released. These MRC datasets focus on a variety of domains of Chinese understanding, such as fact extraction [8, 7], dialogue understanding [34], common sense [14], idiom selection [41] and exam questions used in language proficiency tests [34].

However, all these datasets provide particularly limited challenges for the purpose of building MRC models with the same language proficiency as *native* Chinese speakers. There are 3 major dataset

---

*Equal contribution.

limitations. First, the length of reading materials are **short**. For example, $C_M^3$ [34], a multiple-choice MRC dataset with the longest documents, has merely 180 characters per document on average. Even in the cloze-based datasets, the longest average document length is just around 500 characters. Second, the questions are **not sufficiently difficult**. Most existing datasets are either extractive or domain-specific (e.g., focusing on idiom or simple facts). Although $C^3$ [34] provides exam-based free-form multiple-choice questions, they are designed for *non-native* speakers and therefore do not require native-level reasoning capabilities and common sense knowledge to answer the questions. More importantly, none of existing datasets consider reading comprehension on **classical Chinese** documents, such as classical literature and poetry. Classical Chinese, as a writing style used in almost all formal writing until early 20th century [38], plays a critical role in Chinese culture and has led to numerous idioms and proverbs. Even today, classical literature and poetry are still widely taught and examined in China's education system.

We developed a new general-form multiple-choice Chinese MRC dataset, *Native Chinese Reader (NCR)*, towards building a *native-level* Chinese comprehension system. The NCR dataset contains 8390 documents with over 20K questions collected using the exam questions for the Chinese course in China's high schools, which are designed to evaluate the language proficiency of *native* Chinese youth. Therefore, NCR naturally overcomes the limitations of existing datasets with sufficiently challenging questions and long documents in an average length of 1024 characters over both modern and classical Chinese writing styles (see Table 1).

We provided in-depth analysis of NCR and implemented baselines using popular pretrained models. To further examine the limit of current MRC methods, we additionally launched a online competition using NCR. The best model we obtained achieves an average test accuracy of 59%, which is far below human evaluation result of 79% accuracy. This suggests a significant gap between current MRC model capabilities and the native-level Chinese language proficiency. We hope that NCR could serve as a milestone for the community to benefit future breakthroughs in Chinese natural language understanding.

## 2 Related Work

**English Datasets:** Machine Reading Comprehension tasks require a machine to answer a question based on the content in the given document. Early MRC datasets are primarily *cloze/span-based*, where the answer is simply a span in the document or a few words to be filled in the blank, including CNN/Daily Mail [15], LAMBADA [25], CBT[16], BookTest [2], Who-did-What [24] and CLOTH [39]. The famous SQuAD dataset [27, 26] for the first time introduces human-generated *free-form questions*, which requires the machine to understand natural language to select the correct span in Wikipedia pages. Similar datasets follow this trend of using free-form questions and adopt reading documents from a variety of sources, such as news articles [37, 17] and dialogues [18, 28, 4]. In addition to these datasets where the answers can be directly *extracted* from the document, another popular type of datasets, i.e., *abstractive* datasets, ask the reader to generate an answer that may not be found in the given context [23, 12]. Abstractive datasets further require the reader to perform non-trivial reasoning over the facts in the document as well as common sense knowledge to produce answers. However, since the answer itself is a natural language, evaluation for abstractive datasets can be tricky. *Multiple-choice* datasets overcome the evaluation difficulty in abstractive datasets by simply asking the reader to select the correct answer from the candidate options. Representative datasets, such as RACE [19] and DREAM [33], utilize exam questions collected from standard English proficiency tests, which are generated by language teachers to evaluate a variety of language capabilities of non-native English speakers.

**Chinese Datasets:** The development of Chinese MRC datasets follow a similar trend of English ones. Early cloze-based datasets, such as People Daily news (PD) dataset and Children's Fairy Tale (CFT) dataset [9], utilize a sentence with a repeated noun removed as the question and ask the reader to predict the removed noun. As Chinese counterparts to the SQuAD dataset, DRCD [31], CMRC2017 [8] and CMRC2018 [7] datasets adopt human-generated questions and ask the reader to extract spans in the given documents as answers. DuReader [14], a representative abstractive dataset collects natural questions and answers from Baidu search data, which are in the same style as the English MS-MARCO dataset [23]. CMRC2019 dataset [10] and ChID dataset [41] combine cloze-based questions and multiple-choice answer options. In CMRC2019, a few sentences are

Table 1: Comparison between NCR and related Chinese MRC datasets. NQ is short for free-form *natural question*.

| Dataset | #Que. | Source of Doc. | Que. type | Ans. Type | Doc. # Token Avg. | Classical Chinese |
|---|---|---|---|---|---|---|
| PD | 877K | News | cloze | extractive | 379 | No |
| CFT | 3.5K | Stories | cloze | extractive | 139 | No |
| DRCD | 34K | Wiki | NQ | extractive | 437 | No |
| CMRC 2017 | 364K | Wiki | NQ | extractive | 486 | No |
| CMRC 2018 | 18K | Wiki | NQ | extractive | 508 | No |
| DuReader | 200K | Baidu | NQ | abstractive | 82 | No |
| CMRC 2019 | 100K | Story | cloze | multiple choice | 557 | No |
| ChID | 729K | News&Stories | cloze | multiple choice | 159 | No |
| C3-D | 9.6K | Exam (Non-Native) | NQ | multiple choice | 76 | No |
| C3-M | 10K | Exam (Non-Native) | NQ | multiple choice | 180 | No |
| NCR | 20.4K | Exam (Native) | NQ | multiple choice | 1024 | Included |

masked in each document and the reader is asked to match each option sentence to the corresponding blank in the document. ChID focuses on traditional Chinese idioms by asking the reader to select the correct idiom based on the given story context. Recently, the $C^3$ dataset [34] was released, which contains both free-form questions and multiple-choice answer options. $C^3$ is collected using the exam questions for Chinese-as-a-second-language tests and consists of two sub-datasets, $C_D^3$ focusing on normal documents and $C_M^3$ on dialogues, which can be viewed as the Chinese counterparts of RACE [19] and DREAM [33] respectively.

**Position of NCR:** We developed Native Chinese Reader (NCR), a exam-question-based MRC dataset with free-from questions and multiple-choice answer options, which aims to push the frontier of building *native-level* Chinese MRC models. The high-level statistics of NCR and all the aforementioned datasets are summarized in Table. 1. $C^3$ is perhaps the most related work to ours. However, $C^3$ are collected from *Chinese-as-a-second-language* tests, so its questions are much easier than NCR for three reasons. First, documents, questions and answers in NCR are substantially **longer** than $C^3$. Second, a quarter of the documents in NCR are written in **classical Chinese**, which is a critical component of Chinese language but largely ignored by existing works. We remark that although the answers in ChID dataset [41] are idioms, which is a restricted form of classical Chinese, the documents in ChID remain in modern Chinese. Lastly, the questions in NCR are collected from the exams for China's high-school students and require **native-level reasoning capabilities** using the background knowledge of Chinese history and culture. In-depth comparisons on the question types between $C^3$ and NCR can be found in Sec. 3.4 with example questions shown in Table. 6. In addition, we highlight that a lot of questions in NCR require choosing one *incorrect* option out of 4 options (i.e., three other are *correct*; see Table 4 and 6 for examples). We count the questions containing "不正确" ("incorrect"), "不符合" ("incompatible") or "不恰当" ("inappropriate"), 56.49%, 57.63%, and 56.14% of questions fall into this category in training/validation/test sets respectively. This requires the capability of understanding and reasoning with negations.

Finally, we remark that, in addition to Chinese and English, there are also other datasets developed in other languages like Japanese [32, 36], Russian [13] and cross-lingual scenarios [1, 21], which are of parallel interest to our project.

## 3   Native Chinese Reader (NCR) Dataset

In this section, we provide detailed analysis of the documents and questions in our NCR dataset, including overall statistics, document styles, major challenges as well as studies on question types.

### 3.1   Task Format and Collection Methodology

In NCR, each document is associated with a series of multiple-choice questions. Each question has 2 to 4 options, of which exactly one is correct. The task is to select the correct option based on the document and the question. Both questions and options are expressed in natural language covering a wide range of question types (more details discussed below).

All the questions and documents are collected from online open-access high-school education materials. After data cleaning, 8315 documents followed by 20284 questions are obtained. We randomly split the dataset at the document level, with 6315 for training, 1000 for validation and 1000 for testing. Furthermore, to make sure our test set has sufficient novel questions that never appear online, we also invited a few high-school Chinese teachers to manually generate 193 questions for a total of 73 additional documents to augment the test set. Finally, NCR consists of 6315 documents with 15419 questions for training, 1000 documents with 2443 questions for validation and 1073 documents with 2615 questions for testing.

Table 2: The overall statistics of different Chinese multi-choice MRC datasets. ChID and CMRC2019 are cloze-based without questions. * means statistics are collected over validation and test set only.

| Datasets | Len. of Doc. Avg./Max. | Len. of Que. Avg./Max. | Len. of Opt. Avg./Max. | #Opt. per Que. Min./Avg./Max. | #Que. per Doc. Min./Avg./Max. |
|---|---|---|---|---|---|
| **ChID** | 159.1 / 581 | N/A | 4 / 4 | 7 / 7.0 / 7 | 1 / 1.2 / 12 |
| **CMRC 2019** | 557.3 / 688 | N/A | 13.8 / 29 | 5 / **10.6** / 15 | 0 / **9.9** / 15 |
| **C3-M** | 180.2 / 1,274 | 13.5 / 57 | 6.5 / 45 | 2 / 3.7 / 4 | 1 / 1.9 / 6 |
| **C3-D** | 76.3 / 1,540 | 10.9 / 34 | 4.4 / 31 | 3 / 3.8 / 4 | 1 / 1.2 / 6 |
| **C3** | 116.9 / 1,540 | 12.2 / 57 | 5.5 / 45 | 2 / 3.8 / 4 | 1 / 1.5 / 6 |
| **NCR** Classical only* | 521.5 / 1,258 | 25.7 / 178 | 36.8 / 130 | 2 / 4.0 / 4 | 1 / 2.2 / 5 |
| **NCR** Modern only* | 1207.8 / 4,640 | 24.4 / 276 | 44.1 / 152 | 2 / 4.0 / 4 | 1 / 2.5 / 5 |
| **NCR** All | **1023.7 / 4,640** | **24.5 / 352** | **43.0 / 256** | 2 / 4.0 / 4 | 1 / 2.4 / 5 |

### 3.2 Dataset Statistics

We summarize the high-level statistics of our NCR dataset and other related multi-choice Chinese MCR datasets in Table 2. In addition, we also measure the statistics of classical and modern documents from the validation and test set, where we can observe that modern Chinese articles are more than twice longer than classical Chinese literature. Comparing with other Chinese MRC datasets, NCR is an order of magnitude longer, even including those very concise classical Chinese documents. Besides documents, NCR also contains much longer questions and answer options. Particularly for the option length, NCR is almost an order of magnitude longer except the CMRC2019 dataset. We remark that CMRC2019 is a cloze-style dataset with a completely different question style from NCR: CMRC2019 options are original document texts while the reader only needs to match the options to the corresponding blank in the document. Overall, NCR has substantially longer articles, questions and options with diverse document styles, which suggests a much higher comprehension difficulty than existing datasets.

Table 3: Statistics of document length over NCR validation set and test set. Classical Chinese articles (including poetry) are much shorter than modern Chinese articles.

| Style | count | min | avg | max |
|---|---|---|---|---|
| Modern | 1493 | 47 | 1208 | 4640 |
| Classical | 580 | 24 | 522 | 1258 |
| Poetry | 63 | 24 | 156 | 668 |

### 3.3 Document Style and Challenges

We manually annotated the writing styles of the documents in validation set and test set with summarized statistics in Table 3. Almost a quarter of the documents are in classical Chinese. We remark that most documents are collected from online open-access resources. This indicates that classical Chinese indeed plays a critical role in China's Chinese class, which, however, is often ignored in previous Chinese MRC studies. Table 4 presents two example documents, one in classical Chinese (**D1**) and one in modern Chinese (**D2**), with associated questions. In the following content, we will discuss the major challenges in NCR with respect to different document writing styles.

**Classical Chinese:**    Classical Chinese literature is substantially more difficult than modern Chinese documents due to its conciseness and flexible grammar. Most classical Chinese words are expressed in a single character and therefore are not restrictively categorized into parts of speech: nouns

Table 4: Example documents and questions (left) with English translation (right). Top (**D1**): a classical Chinese poem; Bottom (**D2**): an except of a modern Chinese article. ⋆ denotes the correct option for each question (**Q**).

| | |
|---|---|
| **D1** 相见欢 李煜
无言独上西楼，月如钩。寂寞梧桐深院锁清秋。剪不断，理还乱，是离愁。别是一般滋味在心头。 | **D1** *Form of Xiang-Jian-Huan* **Li Yu**
Silent, solitary, I step up the western tower. The moon appears like a hook. The lone parasol tree locks the clear autumn in the deep courtyard. What cannot be cut nor raveled, is the sorrow of separation: Nothing tastes like that to the heart. |
| **Q1** "寂寞梧桐深院锁清秋"中"锁"的意思是

 A.锁头 B.金锁 **C.锁住** ⋆ D.开锁
**Q2** 下面这首词的赏析不正确的一项是
A. 上阕定景，西楼、月色、梧桐、深院、清秋，画面冷寂。

B. "寂寞梧桐深院锁清秋"一句，写载着梧桐树的院落很寂静，渲染了清秋气氛。

C. 下阕转入抒怀，写出了作者隐忧生活中难以排遣的感情⋆
D. 全词将抽象的情感加以形象化，抒发了作者离乡去国之苦。 | **Q1** In "The lone parasol tree locks the clear autumn in the deep courtyard.", "Locks" means
A.The lock B.Gold lock **C. Lock up** ⋆ D.Unlock
**Q2** The incorrect option for the appreciation and analysis of this poem is
A.The scenery is fixed in the first half, including the west tower, moonlight, parasol tree, deep courtyard, and clear autumn, the picture of which is cold and quiet.
B. The sentence "The lone parasol tree locks the clear autumn in the deep courtyard" says that the courtyard with the parasol tree is very quiet, rendering the atmosphere of autumn.
**C. The second half turns to express feelings and writes about author's unrelievable feeling when he secretly worry about life.**⋆
D. The whole poem visualizes abstract emotions and expresses the author's suffering in leaving his hometown and the capital. |
| **D2** 在酒楼上（节选）鲁迅 ...(2) 我竟不料在这里意外的遇见朋友了，——假如他现在还许我称他为朋友。那上来的分明是我的旧同窗，也是做教员时代的旧同事，面貌虽然顾有些改变，但一见也就认识，独有行动却变得格外迂缓，很不像当年敏捷精悍的吕纬甫了。(11) "我一回来，就想到我可笑。"他一手擎着烟卷，一只手扶着酒杯，似笑非笑的向我说。"我在少年时，看见蜂子或蝇子停在一个地方，给什么来一吓，即刻飞去了，但是飞了一个小圈子，便又回来停在原地点，便以为这实在很可笑，也可怜。可不料现在我自己也飞回来了，不过绕了点小圈子。又不料你也回来了。你不能飞得更远些么？" (20) "你教的是'子曰诗云'"么？我觉得奇异，便问。(21) "自然。你还以为教的是ABCD么？我先是两个学生，一个读《诗经》，一个读《孟子》。新近又添了一个，女的，读《女儿经》。连算学也不教，不是我不教，他们不要教。"(22) "我实在料不到你倒会教这类书，..."(23) "他们的老子要他们读这些；我是别人，无乎不可的。这些无聊的事算什么？只要随随便便，..."(24) "那么，你以后豫备怎么办呢？" (25) "以后？——我不知道。你看我们那时豫想的事可有一件如意？我现在什么也不知道，连明天怎样也不知道，连后一分..." | **D2**. *In the Restaurant* (Excerpt) **Lu Xun** ...(2) I never guessed that here of all places I should expectedly meet a friend – if such he would still let me call him. The newcomer was an old class mate who had been my colleague when I was a teacher, and although he had changed a great deal I knew him as soon as saw him. Only he had become much slower in his movements, very unlike the nimble and active Lu Wei-fu of the old days. (11)"As soon as I came back I knew I was a fool". Holding his cigarette in one hand and the wine cup in the other, he spoke with a bitter smile. " When I was young, I saw the way bees or flies stopped in one place. If they were frightened they would fly but after flying in a small circle they would come back again to stop in the same place; and I thought this really very foolish, as well as pathetic. But I didn't think that I would fly back myself, after only flying in a small circle. And I didn't think you would come back either. Couldn't you have flown a little further?" (20)"Are you teaching that?" I asked in astonishment (21)"Of course. Did you think I was teaching English? First I had two pupils, one studying the Book of Songs, the other Mencius. Recently I have got another, a girl, who is studying the Canon for Girls. I don't even teach mathematics; not that I wouldn't teach it, but they don't want it taught." (22)"I could really never have guessed that you would be teaching such books" (23) "Their father wants them to study these. I'm an outsider so it's all the same to me. Who cares about such futile affairs anyway There's no need to take them seriously ..." (24) "Then what do you mean to do in future?" (25) "In future? I don't know. Just think: Has any single thing turned out as we hoped of all we planned in the past? I'm not sure of anything now, not even of what I will do tomorrow, or even of the next minute ..." |
| **Q3** 下列对文章思想内容的理解与分析，不正确的一项是
A. "行动却变得格外迂缓，很不像当年敏捷精悍的吕纬甫了"高度概括了眼前吕纬甫的精神状态，突出他的迂缓颓废。
B. "蝇子飞了一个小圈子，便又回来停在原地点"，吕纬甫的这番自述自嘲中对自身缺乏清醒的认识，浑霭度日，揭示了残酷的现实生活将人的灵魂挤扁，人们只能在颓唐消沉中磨蚀生命的主题。⋆
C. 从吕纬甫叙述现在教书生涯的内容和原因的话语中，可见他已经违背了当初的理想，变得苟且偷安，屈从于当前顽固封建势力。
D. 文章通过吕纬甫的人生经历来告诉读者，吕纬甫的人生悲剧正是那个时代无数知识分子悲剧命运的代表，而个人的悲剧背后则是整个时代的悲哀。 | **Q3** The incorrect one from the following understanding and analysis of the thought content of the article is:
A. " Only he had becomemuch slower in his movements, very unlike the nimble and active LuWei-fu of the old days. " gives a high-level overview of Lu Weifu's mental state, highlighting his sluggish decadence.
**B. "If they were frightened they would fly but after flying in a small circle they would come back again to stop in the same place." indicates that Lu Wei-fu lacks clear understanding of himself and lives in a muddle, which reveals that the cruel reality of life squeezes the soul, human can only wear out their lives in depression.** ⋆
C. From Lu Wei-fu's narration of the content and reasons of his current teaching career, it can be seen that he has violated his original ideals, has become stubborn, and yielded to the current stubborn feudal forces.
D. The article tells readers through Lu Wei-fu's life experience that Lu Weifu's life tragedy is the representative of the tragedy of countless intellectuals in that era, and behind the personal tragedy is the tragedy of the entire era. |

can be used as verbs, adjectives can be used as nouns, and so on. For example, the character "东" only means "east" in modern Chinese. However, in the classical Chinese sentence, "顺流而东也"(advance eastward along the river), it actually means "advance eastward". Classical Chinese also has distinguishing sentence patterns from nowadays, such as changing the order of characters and often dropping subjects and objects when a reference to them is understood.

Furthermore, an important sub-category in classical Chinese is *poetry*, which is typified by certain traditional poetic forms and rhythms. About 10% of the classical documents in NCR are poetry. Table 4, **D1** shows a famous classical Chinese poem from Song dynasty, which is particularly short

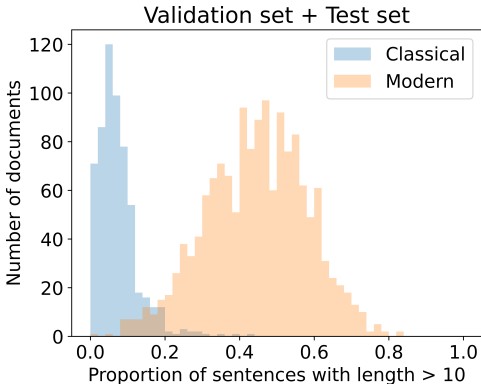
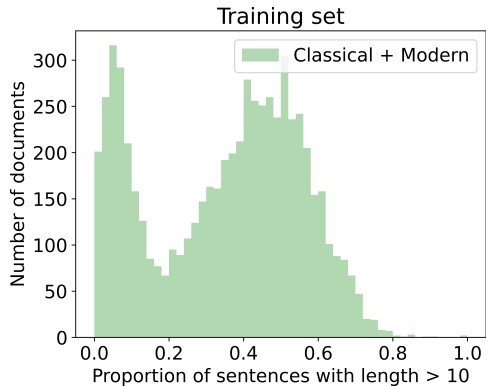

Figure 1: The histogram of $p_{(s>10)}$ over the validation and test sets. The blue bars correspond to classical documents, while the orange bars represent modern documents. We can observe that 0.2 is an effective classification boundary to distinguish classical and modern documents.

Figure 2: The histogram of $p_{(s>10)}$ over the training set. Although the writing style of these documents are not annotated, we could still observe two modalities with a separation boundary at 0.2, which we will use as a rough criterion to distinguish modern and classical documents in the training set.

and abstract in order to satisfy the poetic form of "相见欢" (Xiang-Jian-Huan). This poem describes a scene where the poet stands on a tower staring at the moon. However, in order to correctly understand the sentiment and meaning of the poem, the reader needs to leverage imagery and symbolism in classical Chinese culture (e.g., moon and autumn mean sadness) as well as the personal background of the poet (e.g., Li Yu was a captured emperor).

**Modern Chinese:** For the modern Chinese documents in NCR, in addition to the challenge due to longer average length, the associated questions also focus more on the high-level metaphors and the underlying thoughts, which often require non-trivial reasoning with historical and cultural knowledge. Table 4 **D2** shows an excerpt from a long article (the full document in NCR has about 2000 characters) written by a famous Chinese author, 鲁迅 (Lu Xun). The article describes a scene where the author unexpectedly met one of his old friends not seen for a long time and had a meal together. The associated question (Q3) primarily asks about the high-level thoughts expressed by the author, which has to be inferred from the entire article and requires the readers to have strong knowledge of the author's personal experiences and the background of the era.

In addition to human annotation, we found that sentences in classical documents are usually shorter than in modern documents, which can be used as a simple criterion to categorize writing style. In detail, we split each document into sentences, compute the proportion of sentences with a length greater than 10, and denote it as $p_{(s>10)}$. We plot the histogram of $p_{(s>10)}$ in Figure 1 and 2. We remark that in validation and test sets, 98% of the classical documents have $p_{(s>10)} < 0.2$ while 96% of the modern documents have $p_{(s>10)} \geq 0.2$. This suggests an approximate yet effective categorization criterion, i.e., $p_{(s>10)} < 0.2$, for classifying document style over the training set.

### 3.4 Question Type

To perform fine-grained analysis of the questions in NCR, we conduct human annotations for a sampled batch of 300 questions from the test set. The label of each question is on the consensus of 3 annotators. The questions are categorized into 5 different categories:

**Matching** questions ask about a fact that has been explicitly described in the document. The correct answer can be directly obtained from a short span or a single sentence from the document. Note that different options can refer to different spans.

**Semantic** questions ask about the semantic meanings of words or characters in a sentence, including antonym, synonymy, rhetoric and word segmentation. Q1 in Table 4 belongs to this category. We find that semantic questions are usually associated with classical Chinese documents.

**Summary** questions require the readers to understand all the facts stated throughout the entire document in order to choose a desired option, which presents a correct or incorrect fact summary.

**Reasoning** questions require the reader to perform non-trivial reasoning to infer a conclusion not explicitly stated in the document. A reasoning question in NCR often requires the reader to strong background knowledge and common sense. Q3 in Table 4 and Q1 in Table 6 belong to this category.

**Sentiment** questions ask about the implicit sentiment that the author expressed in the document. Sentiment questions in NCR typically require knowledge of imagery, symbolism and even the author's sociopolitical perspective. Q2 in Table 4 belongs to this category.

The annotations are summarized in Table 5. We can observe that NCR has very few matching questions, which indicates that most NCR questions require non-trivial comprehension of the documents.

As a comparison, we also sampled a total of 600 questions from $C^3$ dataset, another exam-question-based MRC dataset, with 300 from $C_M^3$ and 300 from $C_D^3$ respectively, and annotated the sampled questions with the same standard and annotation process. The statistics are summarized in Table 5. We can observe that $C^3$ contains a large portion of matching questions and much fewer summary and sentiment questions. Despite the fact that NCR and $C^3$ has about the same percentage of reasoning questions, we remark that reasoning questions in NCR are significantly harder than those in $C^3$. This is not only because the documents in NCR are longer (so that fact extraction will be harder) but also because the reasoning questions in NCR typically require reasoning over a combination of document-level facts and background knowledge of both Chinese history and culture. To better illustrate the difference between NCR and $C^3$, we select two example reasoning questions in Table 6, with one from $C^3$ and one from NCR respective.

## 4 Experiment

In this section, we conduct quantitative study as well as human evaluation on our NCR dataset.

### 4.1 Baseline Methods

**Trivial Baselines:** We consider random guess and deterministic choice as trivial baselines. Deterministic choice always selects the same option ID.

**Fine-Tuning of Pretrained Model:** We utilize the MRC model architecture from the BERT paper [11] and perform fine-tuning with NCR. We consider the fine-tuned performance of 7 popular Chinese pretrained models, including BERT-Chinese [11], ERNIE [35], BERT-wwm, BERT-wwm-ext, RoBERTa-large-Chinese [6], MacBERT-base and MacBERT-large [5]. We also investigate the effectiveness of data augmentation by additionally collecting 6000 documents and 13K exam questions for China's primary-school Chinese course. We combine these primary-school exam questions and the NCR training data as an augmented dataset to further boost the final performance. All the model and training details can be found in appendix B.

**Competition:** To examine the limit of current MRC methods, we organized a 3-month-long online competition using NCR with training and validation set released. Participants are allowed to use any open-access pretrained model or any open-access *unlabeled* data. Use of any external MRC *supervision* is forbidden, since a portion of the test questions are possibly accessible online. This aims to prevent human annotations overlapping with our held-out data for a fair competition. There are a total of 141 participating teams and the best submission model with the highest test accuracy is taken

Table 5: Distribution of question types in NCR and $C^3$.

| Type | NCR | $C_M^3$ | $C_D^3$ | $C^3$ |
|---|---|---|---|---|
| Matching | 0.33% | 47.0% | 48.6% | 47.8% |
| Semantic | 20.0% | 1.7% | 2.3% | 2.0% |
| Summary | 38.3% | 15.3% | 4.3% | 9.8% |
| Reasoning | 28.3% | 29.3% | 36.7% | 33.0% |
| Sentiment | 13.0% | 6.7% | 8.0 % | 7.4% |

Table 6: Examples of reasoning questions from NCR (top) and C³ (bottom) with Chinese (left) and English translation (right). We defer the NCR document to Table 2 in Appendix D. ⋆ denotes the correct option. In order to correctly answer Q1 from NCR, the reader not only needs to comprehend the texts in D1 describing the scene where Jun-tu meets the author's mother but also needs to understand the cultural meaning of "老太太" (madam).

| | | |
|---|---|---|
| NCR | D1：故乡（节选）鲁迅 | D1: *My old home* (excerpt) **Lu Xun** |
| | Q1：选出与本选段中中年闰土形象分析不恰当的一项 | Q1: Select the incorrect analysis of middle-aged Jun-tu's image. |
| | A. 他称"我"母亲为"老太太"，表现了他有意讨好"我"母亲。⋆ | **A.He called my mother the "madam", which shows his intention to please my mother.** ⋆ |
| | B.他称自己少年时的好友为"老爷"，说明了他受封建等级观念影响很深。 | B. He called his former good friend "master", which shows that he was deeply influenced by the feudal concept of hierarchy. |
| | C.从他的对话中可以看出他的生活景况非常不好，他是当时下层人民形象的缩影。 | C. From his dialogue, we can see that the situation of his life is very bad, and he is the epitome of the image of the lower class at that time. |
| | D.宏儿和水生就像当年的"我"和闰土一样，彼此之间没有隔阂。 | D. Hung-eth and the Shui-sheng are just like me and Jun-tu at that time, they are not estranged from each other. |
| C³ | D2：   男：还能不能再便宜点儿？" 女："已经给您打五折了，先生！" | D2. Man: Can you make it a little cheaper? " Woman: "I've given you a 50% discount, sir!" |
| | Q2：他们最可能是什么关系？ | Q2. What is the most likely relationship between them? |
| | A.夫妻   C.老师和学生 | A. Husband and wife    C. Teacher and student |
| | B.同事   D.售货员和顾客 ⋆ | B. Colleagues    **D. Salesperson and customer** ⋆ |

as the *competition* model. The team is from an industry lab. They first pre-trained an XLNet-based model [40] on a company-collected large corpus[2]. For each question, they use an information retrieval tool Okapi BM25 [30, 29] to extract the most relevant parts from the document and then run this pre-trained model for answer selection based on the extracted texts.

**Human Evaluation:** We randomly sample 50 documents with 120 questions from the annotated subset of test data in NCR and send these questions to 30 sophomore college students. All the students are native Chinese speakers majored in computer science, who have not taken any Chinese course during the recent 2 years after college admission. Therefore, we believe they have reasonable reading comprehension capabilities close to typical China's high school students. Each question is completed by at least 3 students to get an accurate performance estimation.

## 4.2 Results

### 4.2.1 Overall Performance

The performance of different baseline methods as well as human volunteers are summarized in Table 7. Pretrained models are substantially better than trivial baselines. Particularly, the MacBERT-large model produces the highest fine-tuning test accuracy of 0.4780 while the use of external data augmentation further boost the test performance to 0.5021, which suggest the effectiveness of data augmentation. The best model comes from the participants of the online competition. The best competition model achieves a test accuracy of 0.5985, which is much higher than the best fine-tuning model that authors obtained. However, the human volunteers achieves an average accuracy of 0.7917, which results in a 20% performance margin over the best baseline MRC model. Human performance is averaged per question over three annotators. To measure the inter-agreement, we calculate the agreement ratios. 60.83% of the questions have the same answer from all the 3 students, and 96.67% of the questions have the same answers from at least 2 students.

---

[2]Unfortunately, the company disagreed to release their internal pretraining data but the final trained model will be released at our project website.

Table 7: Validation and test accuracy of different MRC methods on NCR. * Human evaluation is only conducted over a subset of test questions.

| Method | Val. | Test |
|---|---|---|
| Random Guess | 0.2505 | 0.2511 |
| Deterministic Choice | 0.2951 | 0.2613 |
| BERT-Chinese | 0.3930 | 0.3946 |
| ERNIE | 0.4445 | 0.4252 |
| BERT-wwm | 0.4310 | 0.4272 |
| BERT-wwm-ext | 0.4814 | 0.4451 |
| MacBERT | 0.4736 | 0.4597 |
| RoBERTa-large-Chinese | 0.4666 | 0.4642 |
| MacBERT-large | 0.5051 | 0.4780 |
| MacBERT-large (data aug.) | **0.5199** | **0.5021** |
| Competition | 0.5831 | 0.5985 |
| Human performance | N/A | 0.7917* |

Table 8: Test accuracy of human and AI w.r.t. different document writing styles. FT is the best model finetuned by ourselves and CMP is the best competition model.

| Document Style | Human | FT | CMP |
|---|---|---|---|
| Modern | 0.7489 | 0.5257 | 0.6151 |
| Classical (w/o poetry) | 0.8632 | 0.4502 | 0.5671 |
| Poetry only | 0.9167 | 0.3462 | 0.4179 |

### 4.2.2 Fine-Grained Analysis

We measure the performance of the MRC models, i.e., the best fine-tuned model (**FT**) and the competition model (**CMP**), and human on the test questions w.r.t. different factors, including writing style, document length and question type. We remark that the model accuracy are measured over the entire test set except the study on question type, which are over the annotated subset only.

**Writing style:** Table 8 illustrates the performance of MRC models and human on different document styles. The performance of AI significantly drops on classical Chinese documents, particularly on poetry. By contrast, we observe an opposite phenomena for humans, who perform the best on poetry, the most abstract form of classical Chinese literature. We remark that in China's Chinese exams, questions on modern and classical texts may often have different examination focuses. Questions on classical Chinese are more biased towards understanding the meaning of characters, words, and sentences (see Table 4, Q1), which may not be intrinsically difficult for native Chinese students who have been well trained. While modern Chinese questions are often more general and require in-depth understanding of the entire document (see Table 4, Q3), which can be challenging for humans. This is because Chinese documents are much longer and reading and remembering facts under long documents can easily make a human distracted.

We also investigate whether a AI model purely trained on modern Chinese can directly transfer to classical texts. Since we do not have ground truth annotations on training set, we following the filtering process in section 3.3 as a rough categorization, which yields 4507 documents. We fine-tune the MacBERT-large on this filtered training set (without data augmentation) and show the testing results in Table 9. We can observe that the performance on classical documents drops significantly with classical documents filtered out, while the performance on modern documents remains unchanged. Hence, we argue that classical Chinese training data can be critical. This also suggests an important direction for improving Chinese pre-trained models.

**Document Length:** Table. 10 summarizes the performance of human and AI on documents with various lengths. For classical Chinese documents, the most challenging documents are those shortest ones, which are most likely poetry. For modern articles, human performance drops for particularly long articles. For AI, the hierarchically-structured competition model performs the worst on relatively short articles while the fine-tuning model has the most difficulties in documents with a moderate length, i.e., from 300 to 600 characters. This suggests possible enhancements on model architecture.

Table 9: Test accuracy of MacBERT-large fine-tuned on complete and filtered training set.

| Training set | Modern | Classical (w/o poetry) | poetry |
|---|---|---|---|
| Complete | 0.5039 | 0.3871 | 0.3582 |
| Filtered | 0.5060 | 0.2970 | 0.2836 |

Table 10: Test accuracy of human and AI w.r.t. different document lengths in both classical and modern Chinese. Human data are only presented when at least 5 documents can be collected from the annotated subset.

| Classical | | | | |
|---|---|---|---|---|
| Len. | [0,100] | (100, 300] | (300, 600] | >600 |
| FT | 0.3014 | 0.5505 | 0.4162 | 0.4203 |
| CMP | 0.2192 | 0.5046 | 0.6069 | 0.6116 |
| Human | N/A | 0.8333 | 0.7333 | 0.8958 |
| Modern | | | | |
| Len. | [0,300] | (300, 600] | (600, 1200] | > 1200 |
| FT | 0.5714 | 0.4203 | 0.5220 | 0.5486 |
| CMP | 0.3809 | 0.5652 | 0.6373 | 0.5973 |
| Human | N/A | N/A | 0.7879 | 0.6970 |

Moreover, although the performance gap between human and machine becomes less significant on particularly long documents, the accuracy of AI systems remains unsatisfying in general. We also want to remark that even those relative short documents in NCR are substantially longer and more sophisticated than existing datasets of the similar question types like $C^3$.

Table 11: Test accuracy of human and AI w.r.t. different question types. We ignore matching questions since they are too infrequent.

| Question Type | Human | FT | CMP |
|---|---|---|---|
| Semantic | 0.9047 | 0.5000 | 0.5833 |
| Summary | 0.7976 | 0.5431 | 0.5603 |
| Reasoning | 0.7179 | 0.6000 | 0.6588 |
| Sentiment | 0.6333 | 0.5641 | 0.5128 |

**Question Type:** We also compare the performance of human and AI on different question types in Table. 11. To our surprise, sentiment questions, which are the most challenging for humans, yield the smallest performance margin. While the largest gap is on semantic questions, which we believe the easiest for human. This indicates that a pretrained model is capable of capturing high-level sentiment information but still lacks word/character-level reasoning abilities. In addition, we also observed that the hierarchical competition model performs much worse than the fine-tuned model on sentiment questions, which suggests that running a retrieval model first may result in a loss of document-level global information which can be critical for sentiment analysis. This raises an open challenge for building more effective hierarchical models for processing long texts.

## 5 Conclusion

We present a novel Chinese MRC dataset, *Native Chinese Reader (NCR)*, towards building *native-level* Chinese MRC models. Experiments on NCR indicate a significant gap between current MRC methods and human performance, which suggests great opportunities for future research, and, hopefully, pushes the frontier of Chinese natural language understanding.

**Remark:** Our dataset primarily consists of open-access exam questions or generated ones with teacher permission. All the documents are all public teaching materials. The released models are permitted by the online competition participants. Annotations and human evaluation results are completed by PhD students and interns that are all paid according to our institute regulation. Hence, we believe that our project will not lead to any legal or ethical issues.

**Acknowledgements**

Yi Wu is supported by 2030 Innovation Megaprojects of China (Programme on New Generation Artificial Intelligence) Grant No. 2021AAA0150000. We would also like to thank the anonymous reviewers for their insightful feedbacks.

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
