# OpenReview forum: "Native Chinese Reader: A Dataset Towards Native-Level Chinese Machine Reading Comprehension"
_NeurIPS.cc/2021/Track/Datasets_and_Benchmarks/Round2 — NeurIPS 2021 Datasets and Benchmarks Track (Round 2)_

### Official Review · Reviewer_2zwd · 2021-09-02
**An interesting and intellectually challenging dataset on Chinese MRC**

**Rating:** 8
**Confidence:** 4
**Clarity:** The paper is clear and easy to follow.

**Strengths:**

This work curates a dataset, NCR, that is significantly different from and more challenging than existing Chinese MRC datasets (i.e. C3, CMRC2019). The authors show well-rounded statistics that distinguish the new dataset in terms of document length and question types. Solving the new dataset will require holistic passage understanding, deeper reasoning, and incorporating cultural knowledge.

A comprehensive benchmark is set for NCR using various existing Chinese pretrained LMs, along with an exhaustive fine-grained analysis and discussion. Furthermore, the authors did not stop there, but went ahead to organize an open competition that elicited a much stronger benchmark.

**Weaknesses:**

In Section 3.3, the authors should describe how they define the boundary between classical and modern Chinese, and how they instructed the annotators to make such a distinction. A common way is to draw the line at the Chinese renaissance (新文学运动) around 1920, yet some may consider the famous fictions *Journey to the West* (《西游记》) and *Water Margin* (《水浒传》) as written in somewhat modern Chinese despite they were composed hundreds of years ago, while contemporary poets may write poetry in ancient formats and thus also blurs the boundary.

To further show the high quality of the dataset, the authors should probably investigate if there are annotation artifacts and spurious correlations (i.e. certain patterns in the use of language that easily gives away the correct answer). For example, in Table 4 Q3.B, the expression “can only” (“只能”) is an absolute quantifier that usually signals an inappropriate statement. It’d be great to know how well a model can perform by only exploiting this kind of surface patterns in the choices.

**Additional Feedback:**

In lines 118-123, the number of questions is not consistent: 20477+193==15419+2443+2613 ?

It may be worth highlighting that a lot of questions in NCR are postulated in a way that requires choosing the one “incorrect” option out of 4 options (the other three are “correct”), which requires the ability to understand and reason with negations.

In Section 4.1, the authors forbid the usage of additional labeled data in the competition (line 218), yet they use additional training data to help finetune the model (line 214). Are these additional data *labeled*? If so, are these part of the NCR dataset so that competition teams were allowed to use them?

In Section 4.1, did you try fine-tuning your model on only the modern-style part of the NCR dataset? I wonder how well the MRC ability transfers to classical-style documents if it has only seen modern-style data. It may show the importance of having classical-style data in the training set in helping the model understand classical literature.

Tables for experiment results could be improved aesthetically (e.g. using bold fonts, thicker lines, more space).

**Correctness:**

The dataset collection and split seem appropriate. Experiments and benchmark setup also seem correct.

**Documentation:**

Dataset documentation is satisfying. The dataset is hosted on Google Drive and is downloadable.

The code is also hosted on Google Drive as a zipball, which is a bit hard to browse and maintain. I’d recommend hosting the codebase on GitHub or equivalent platforms.

**Ethics:**

Can the authors share the URL to their data source, which seems to be “online open-access high-school education materials” (line 117).

**Relation To Prior Work:**

Prior work is comprehensively summarized, both on English and Chinese MRC datasets. This work has a nice niche that it aims to build a native-level collection of questions, which is inspired by the English MRC trend to collect data from exam questions, while distinguishing itself from prior dataset collection efforts on Chinese MRC.

**Summary And Contributions:**

The paper presents NCR, a large-scale multiple-choice Chinese MRC dataset consisting of high-school-level exam questions. This dataset is unique and of high quality because it contains longer documents in general, requires more in-depth understanding and reasoning, and encompasses both classical and modern writing styles of Chinese.

Benchmarks are set by both the authors’ own efforts and the winner of a public competition. The fine-grained analysis reveals various limitations of current MRC models, pointing out ways for further improvement.

---

> ### Author Response · Authors · 2021-09-29
> **Thank you for the constructive feedback and valuable ideas.**
>
> We thank the reviewer for the constructive feedback and valuable ideas for future research!
>
> **Modern/Classical boundary:** We completely define classical and modern Chinese based on writing style rather than writing age. Since the dataset comes from exams, documents in different writing styles usually aim to examine different capabilities, the boundary in NCR is clear and there is no such ambiguity like Journey to the West and Water Margin when annotating. As for poetry, we treat the poetry in ancient formats as classical-style poetry.
>
> **Surface patterns:** We investigate some special patterns and predict the answer based on these patterns. we focus on several absolute quantifiers including “只能” (“can only”), “必然”, “必定”, “必须”(“must”), “只可能”, (“may only”), “绝对” (“absolute”).  For each pattern, We keep the questions where only one option contains this pattern or only one option doesn’t contain this pattern and choose the special option as the predicted answer. The results are aggregated from the whole dataset with a total of 20477 questions.  “combination” represents a combination pattern that including all the aforementioned patterns. We can observe that there are not many questions meeting the requirements, and the accuracy is indeed higher than random but also still very low.
>
> |Pattern|只能|必然|必定|必须|只可能|绝对|combination|
> |--|--|--|--|--|--|--|--|
> |#Question|291|308|51|658|5|101|1296|
> |Accuracy|0.3470|0.2825|0.3077|0.2903|0|0.2673|0.2994|
>
> **Inconsistent numbers:** Thank you for pointing out the wrong numbers, we double-check the dataset, 20477 is the total number of questions where 193 new questions are already included. And there are 2615 questions in the test set. The correct numbers are 20477== 15419+2443+2615. We have updated the numbers in our new version.
>
> **Reasoning with negations:** Thanks for your suggestions highlighting that a lot of questions require choosing the “incorrect” option.We count the questions containing “不正确” (“incorrect”), “不符合” (“incompatible”) or “不恰当” (“inappropriate”), 56.49%, 57.63%, and 56.14% of questions fall into this category in training/validation/test sets respectively. This requires the capability of understanding and reasoning with negations. We have highlighted it in the new version.
>
> **Additional data:** The additional data we used are labeled and comes from the Chinese course in China’s primary school. Because the quality and difficulty are relatively lower, we didn’t include this data in NCR. But we can make these data public as supplemental material for further research.
>
> **Transfer to classical texts:** The idea of fine-tuning the model on only the modern-style part of the NCR dataset is wonderful. But unfortunately, we didn’t label the training set as classical/modern. To verify this idea, we utilize that sentences in classical documents are usually shorter than in modern documents, which can be used as a simple criterion to categorize writing style. In detail, we split each document into sentences, compute the proportion of sentences with a length greater than 10, and denote it as $p_{(s>10)}$. In the validation and test sets, 98% of the classical documents have $p_{(s>10)} < 0.2$ while 96% of the modern documents have $p_{(s>10)}>0.2$.  This suggests an approximate yet effective categorization criterion, i.e., $p_{(s>10)}<0.2$, for classifying document style over the training set. We filter out the documents with $p_{(s>10)}<0.2$,  which yields 4507 documents.  We fine-tune the MacBERT-large on this filtered training set (without data augmentation) and show the testing results in the following table.  We can observe that the performance on classical documents drops significantly with classical documents filtered out, while the performance on modern documents remains unchanged. Hence, we argue that classical Chinese training data can be critical.
>
> |Training set|Modern|Classical(w/o poetry)|poetry|
> |--|--|--|--|
> |Complete|0.5039|0.3871	|0.3582|
> |Filtered|0.5060|0.2970|0.2836|
>
> Thanks for your suggestions!  we have uploaded the code to Github [https://github.com/xssstory/NCR_baseline](https://). And we have updated our paper to include all the discussions above.

---

### Official Review · Reviewer_eZSy · 2021-09-06
**A Challenging Dataset for Chinese Machine Reading Comprehension.**

**Rating:** 7
**Confidence:** 4
**Correctness:** The data collection and evaluation me…
**Clarity:** The paper is well organized and easy …

**Strengths:**

- This is a meaningful dataset for measuring the progress of current Chinese MRC models.
- The dataset and baseline code is publicly available.
- The dataset contains diverse types of questions, such as matching based questions that ask about a fact that has been explicitly described in the document, summary questions that require the readers to understand all the facts stated throughout the entire document, sentiment questions that ask about the implicit sentiment that the author expressed in the document.





**Weaknesses:**

- There should be a more detailed description of the implementation of baselines. Given the averaged length of documents (1024 tokens) is much larger than the maximum sequence length of language models (512 tokens for BERT), how do you encode these documents? Do you truncate these documents to the maximum sequence length of language models? What is your truncation strategy?
- There could be some error analysis to help readers understand the current limitation of baseline models. What are the common mistakes made by models? How could further work improve them?


**Additional Feedback:**

None

**Documentation:**

The authors have built a dataset website that contains information about the organization, availability and maintenance, and responsible use.

**Ethics:**

I do not find any ethical concerns in this dataset.

**Relation To Prior Work:**

This dataset provides several unique features, including:
- long documents,
- classical Chinese documents,
- diverse types of questions.

**Summary And Contributions:**

This paper proposes a Chinese machine reading comprehension(MRC) dataset which consists of 8315 documents and 20477 questions. It is more challenging compared to previous Chinese MRC datasets because it covers diverse documents, such as classical Chinese literature and modern Chinese articles, and its average length is one order of magnitude longer than the previous dataset. Experimental results show that the best model achieves 59% test accuracy while humans achieve an average accuracy of 79%, which indicates a significant performance gap between current Chinese MRC models and humans.

---

> ### Author Response · Authors · 2021-09-29
> **Thank you for the constructive feedback, questions and suggestions**
>
> We thank the reviewer for the helpful feedback. We address individual points below:
>
> > There should be a more detailed description of the implementation of baselines. Given the averaged length of documents (1024 tokens) is much larger than the maximum sequence length of language models (512 tokens for BERT), how do you encode these documents? Do you truncate these documents to the maximum sequence length of language models? What is your truncation strategy?
>
> We truncated these documents to fix the maximum sequence length of language models by dropping the end part of the documents.
>
> > There could be some error analysis to help readers understand the current limitation of baseline models. What are the common mistakes made by models? How could further work improve them?
>
> We try to analyze the error taken by baseline models, but it is difficult to find some common mistakes. The difficulty mainly comes from that 1) The questions are about multiple choice and there are no more meanings of the model’s outputs. 2) Options of different questions are multifarious.
>
> In replacement, we analyze the accuracy w.r.t. different writing styles,  different document lengths, and different question types in the paper, and hope these analyses could help readers understand the current limitation of baseline models.
>
> In addition, following the suggestion of Reviewer 2zwd, we investigate whether an AI model purely trained on modern Chinese can directly transfer to classical texts. The results on the test set show that the performance on classical-style documents drops significantly, while the performance on modern-style documents remains unchanged. This shows the importance of having classical-style data in the training set in helping the model understand classical literature. This also suggests an important direction for improving Chinese pre-trained models. We have updated these results in the new version.

---

### Official Review · Reviewer_ao4a · 2021-09-14
**A Chinese multi-choice MRC dataset**

**Rating:** 5
**Confidence:** 5
**Correctness:** The construction of the dataset is re…

**Strengths:**

1. The sample quality of NCR is high, as these samples are from real exams.
2. There are various types of text in NCR, making it more challenging.

**Weaknesses:**

1. Apart from NCR's high quality, there is not many new things to learn from this paper, especially for non-Chinese speakers. The collecting process and dataset analyses are quite normal. I am not sure the paper fits NeurIPS.
2. Some of the statements are not in accordance with source codes, and some of the details are missing.
3. The development and test sets are not classified into modern/classical/poetry subsets, which hinders us from understanding the weaknesses in the current Chinese MRC systems.

**Additional Feedback:**

Questions and comments:

1. In human evaluation, it is unclear how human performance was calculated. Is it an average score among all three student annotators? What is the inter-agreement?
2. In Table 8, the human performance is somewhat surprised me. As I suppose the performance of modern document to be the highest. However, it seems that modern documents are the most difficult subset. Do you have any further comments on this? The content in line 247-251 seems to be quite shallow to me.
3. Why are the dev/test sets not classified into modern/classical/poetry categories? It is OK to have an overall score, but we need more precise classifications here. Each category is quite different from others, and thus it is not good to blend them as a whole. After checking the original dataset, the reviewer did not find category information. Without such meta-data, we are not able to analyze how our MRC system performs on each category and adjust the model design for further improvements.
4. I downloaded the dataset and source code for baselines from https://sites.google.com/view/native-chinese-reader. After a rough scanning, it seems that the baselines are based on Longformer, but not pure BERT/RoBERTa. While in both the main paper and appendix, there are no illustrations on this nor citation to Longformer. This causes a discrepancy of your paper and the actual baselines that were used.
5. The test set seems to be already made public, which may affect the integrity of testing process for future works. Why did you not keep it as private (as you have organized a competition beforehand)?

Grammatical issues / Typos / Minor comments:

1. line 126: high level -> high-level
2. line 222: from a industry -> from an industry
3. line 228: token -> taken
4. Table 8: The size of the left and right quote mark differs in '(w/o poetry)'.
5. Please be consistent in showing your evaluation results. For example, in line 236-240, you use percentage for accuracy, while it is not in Table 7/8/9/10.

**Clarity:**

Mostly. But some of the illustrations are not in accordance with the source codes. Please see 'Additional Feedback' for detail.

**Documentation:**

Acceptable but not thorough. Please see NeurIPS CFP for more information.
Quote from CFP:
```
 Submission introducing new datasets must include the following in the supplementary materials (as a separate PDF):
 Dataset documentation and intended uses. Recommended documentation frameworks include datasheets for datasets, dataset nutrition labels, data statements for NLP, and accountability frameworks.
```

For example, use data sheets for datasets: https://arxiv.org/abs/1803.09010

**Ethics:**

None.

**Relation To Prior Work:**

Yes.

**Summary And Contributions:**

In this paper, the authors propose a new machine reading comprehension (MRC) dataset for Chinese, called Native Chinese Reader (NCR). NCR typically follows a multi-choice MRC task form, which is collected from the exams in Chinese high schools. NCR consists of various text genres, such as modern/classical/poetry Chinese text, which makes this dataset more native to Chinese. The resulting dataset contains 20K questions and has an official train/dev/test split. To test the performance, they used BERT/MacBERT for building baselines. The results show that the best model still has a big gap in human performance, indicating that NCR is challenging.

---

> ### Author Response · Authors · 2021-09-29
> **Thank you for the constructive feedback, questions and suggestions**
>
> We thank the reviewer for the careful and helpful review. We address individual points below:
>
> We want to emphasize that there has been little effort in the MRC literature on such challenging reading materials and questions, so we believe NCR could serve as a new testbed to push the frontier of AI systems towards native-speaker-level language understanding capability, which we believe fit the scope of NeurIPS dataset track. Regarding the Chinese language, Chinese, as one of the most widely used languages in the world, is particularly important for AI systems and there have been many published works on Chinese NLP tasks in top conferences every year [1]. Even in this year’s NeurIPS dataset track, there have been NLP dataset papers in other languages, i.e., Romanian, Italy that have been already accepted [2]. Finally, we believe that the technical advances by NCR should also benefit the entire NLP community beyond the Chinese language. We want to remark that, from our bi-lingual examples in the paper, it is clear that similar MRC challenges can be developed in English as well ---- although these types of questions are not covered in the SAT/LSAT exam, which makes the development of a similar English dataset much more expensive than Chinese.
>
> **Human evaluation:** In human evaluation, human performance is an average score among all three students and is computed by (# correct answers / # total questions). For example, there are 120 questions, and each question is completed by 3 students, so # total questions = 360, # correct answers is the number of correct answers of all students. We calculate the ratios of agreement between the answers of 3 students to measure the agreement. 60.83% of the questions have the same answer from all the 3 students, and 96.67% of the questions have the same answers from at least 2 students.
>
> **Human performance:** Since our dataset comes from exams, there are usually different focuses on the inspection of classical text and modern text. Questions about classical Chinese are more biased to examine whether the students can correctly understand characters, words, and sentences (see Table 4, Q1), which are not intrinsically difficult for native Chinese students who have been well trained. But questions in modern Chinese are more general and require a more in-depth understanding of the document (see Table 4, Q3), which can be challenging for humans. Another reason is that the modern Chinese documents are much longer and the options under a single question may cover the entire article. Reading and remembering facts under a long document can easily make a human tired or less concentrated.
>
> **Dataset categories:** We do have classified the dev/test sets in the public datasets. In the dev/test set there is a field named "Type" for each document, where "00" means modern Chinese without poetry, "11" means classical Chinese without poetry, "22" means classical poetry, and "33" means modern poetry. Since there is little modern poetry, we combine "00" and "33" as modern. The corresponding statistics numbers are listed in Table 3 in our paper. We have added this illustration to our website [https://sites.google.com/view/native-chinese-reader/datasheets-for-datasets.](https://)
>
> **Code:** All the results listed in the paper could be reproduced by this code based on pure BERT/RoBERTa/MacBert. Longformer is independent of other codes, we tried Longformer but got worse performance. You can follow the "README" in the code to reproduce our results. Our implementation is based on the packages "transformers" and "PyTorch". To better maintain the code, we have released the code in GitHub [https://github.com/xssstory/NCR_baseline](https://) and removed the part of Longformer to avoid misunderstanding.
>
> **Test set:** We think that it is normal to make the test set public for others who want to further study the dataset, and almost all the datasets release their training/dev/test set[1, 3].  We keep the dataset private to guarantee the authenticity and credibility of the competition. After the completion of the competition, we think that making the datasets public is beneficial to further research.
>
> **Documentation:** We have released the datasheets for datasets and data statements in [https://sites.google.com/view/native-chinese-reader/datasheets-for-datasets](https://) and [https://sites.google.com/view/native-chinese-reader/statement](https://).
>
> Thank you for pointing out other issues and typos, we have updated our paper to fix these issues and the discussions above are also included.
>
> [1] Sun, Kai et al. “Investigating Prior Knowledge for Challenging Chinese Machine Reading Comprehension.” Transactions of the Association for Computational Linguistics 8 (2020): 141-155.
>
> [2] Dumitrescu, Ș. et al. “LiRo: Benchmark and leaderboard for Romanian language tasks.” (2021).
>
> [3] Rajpurkar, Pranav et al. “SQuAD: 100,000+ Questions for Machine Comprehension of Text.” EMNLP (2016).

---

> > ### Author Response · Authors · 2021-09-30
> > **Supplementary materials are updated.**
> >
> > In addition to the website, we also submit a separate PDF of the datasheets in the supplementary materials.

---

### Author Response · Authors · 2021-09-30
**Paper and supplementary are updated**

Given the Reviewers' suggestions, we uploaded an updated version of the paper and the supplementary, with the changes marked in red.

In the main paper, we

1. highlight that a lot of questions in NCR require the capability of understanding and reasoning with negations; (line 106 ~ 110)
2. correct the inconsistent numbers; (line 123, 128)
3. fix the typos; (line 131, 233, 239, 247~250, Table 8)
4. propose a simple criterion to categorize writing style for training data; (line 176 ~182, Figure 1, 2)
5. descript the details of human performance and measure the inter-agreement; (line 252~255)
6. explain the human performance of Table 8; (line 264 ~ 271)
7. investigate whether an AI model purely trained on modern Chinese can directly transfer to classical texts (line 272 ~278, Table 9)

In supplementary materials, we

1. submit a separate PDF of the datasheets;
2. add the truncation details; (Appendix B)
3. investigate some special patterns. (Appendix C)

---

### Decision · Program_Chairs · 2021-10-09

**Decision:**

Accept

**Comment:**

All reviewers agree this paper proposes a new and challenging reading comprehension dataset in Chinese that goes beyond past work in offering high-quality and diverse reading comprehension questions.